

# Are neonicotinoid insecticides driving declines of widespread butterflies?

Andre S. Gilburn[1], Nils Bunnefeld[1], John McVean Wilson[1], Marc S. Botham[2], Tom M. Brereton[3], Richard Fox[3] and Dave Goulson[4]

[1] Biological and Environmental Sciences, University of Stirling, Stirling, Scotland, United Kingdom
[2] Biological Records Centre, CEH Wallingford, Crowmarsh Gifford, Wallingford, Oxfordshire, United Kingdom
[3] Butterfly Conservation, East Lulworth, Wareham, Dorset, United Kingdom
[4] School of Life Sciences, University of Sussex, Brighton, Sussex, United Kingdom

Corresponding author
Andre S. Gilburn,
andre.gilburn@stir.ac.uk

## ABSTRACT

There has been widespread concern that neonicotinoid pesticides may be adversely impacting wild and managed bees for some years, but recently attention has shifted to examining broader effects they may be having on biodiversity. For example in the Netherlands, declines in insectivorous birds are positively associated with levels of neonicotinoid pollution in surface water. In England, the total abundance of widespread butterfly species declined by 58% on farmed land between 2000 and 2009 despite both a doubling in conservation spending in the UK, and predictions that climate change should benefit most species. Here we build models of the UK population indices from 1985 to 2012 for 17 widespread butterfly species that commonly occur at farmland sites. Of the factors we tested, three correlated significantly with butterfly populations. Summer temperature and the index for a species the previous year are both positively associated with butterfly indices. By contrast, the number of hectares of farmland where neonicotinoid pesticides are used is negatively associated with butterfly indices. Indices for 15 of the 17 species show negative associations with neonicotinoid usage. The declines in butterflies have largely occurred in England, where neonicotinoid usage is at its highest. In Scotland, where neonicotinoid usage is comparatively low, butterfly numbers are stable. Further research is needed urgently to show whether there is a causal link between neonicotinoid usage and the decline of widespread butterflies or whether it simply represents a proxy for other environmental factors associated with intensive agriculture.

## INTRODUCTION

In England, the overall population level of widespread butterflies on monitored farmland sites declined by 58% between 2000 and 2009 (*Brereton et al., 2011*). This has occurred despite conservation spending in the UK more than doubling in real terms over the same period (*JNCC, 2015*), with some of this spend funding agri-environment schemes that ought to benefit farmland biodiversity (*Batáry et al., 2015*). Additionally, models predict

that moderate climate change should actually benefit most UK butterflies through warmer summer temperatures (*Roy et al., 2001*; *Warren et al., 2001*). Although habitat deterioration resulting from land-use change (e.g., agricultural intensification) is likely to be the most important factor driving long-term declines of UK butterflies (*Warren et al., 2001*) and weather the principle factor determining inter-annual fluctuations (*Brereton et al., 2011*), the precise reasons for the recent decline are undetermined.

While the negative impact of modern, intensive agriculture on biodiversity has been widely recognised (*Donald, Green & Heath, 2001*; *Kleijn et al., 2009*), the relative contribution that agricultural pesticides make to this overall impact has rarely been examined (*Gibbs, Mackey & Currie, 2009*), and never, to our knowledge, on population-level trends in butterflies. A recent review assessing the impacts of pesticides on non-target species has identified a clear need for studies investigating the effects of pesticides on Lepidoptera that inhabit farmland (*Pisa et al., 2015*). We seek to address this by considering the effect of neonicotinoid insecticides on population trends of widespread UK butterflies. Neonicotinoid insecticides were introduced in the mid-1990s and are now widely used in arable farming both in the UK and globally. They are most commonly used as a seed dressing on oilseed rape and cereals (*Elbert et al., 2008*), with the intention that the active ingredient is absorbed by the seedling and spreads systemically through the crop tissues. Studies into the effects of neonicotinoids on non-target species have mainly focussed on bees as concentrations ranging from 0.6 to 51 parts per billion (ppb) have been found in the nectar and pollen of treated crops (*Cresswell, 2011*; *Blacquière et al., 2012*; *Pisa et al., 2015*). Neonicotinoids have been found to have a range of sublethal effects on honeybees and bumblebees, including impaired navigation and learning, reduced colony growth, impaired immunity and reduced queen production (*Cresswell, 2011*; *Henry et al., 2012*; *Whitehorn et al., 2012*; *Gill, Ramos-Rodriguez & Raine, 2012*; *Di Prisco et al., 2013*), but field experiments in which honeybee colonies have been exposed to plots of treated crops have not found significant impacts on colony health (*Cutler & Scott-Dupree, 2007*; *Pilling et al., 2013*). It remains disputed whether neonicotinods are having significant impacts on bees in field situations (*Godfray et al., 2014*; *Pisa et al., 2015*). However, a recent field study has shown that treated rape seeds reduce the density of wild bees, the nesting of solitary bees and the growth of bumblebee colonies (*Rundlöf et al., 2015*).

Studies investigating the impacts of neonicotinoids on butterflies are completely lacking and urgently required (*Pisa et al., 2015*). Butterflies could be exposed to neonicotinoids via nectar from crops but also through contamination of non-crop plants and habitats. There are two routes by which seed-dressing neonicotinoids could contaminate farmland habitats such as field margin vegetation (*Goulson, 2014*). Firstly, dust produced during drilling of dressed seeds can contain high concentrations of neonicotinoid, which drifts onto surrounding vegetation (*Krupke et al., 2012*; *Girolami et al., 2013*; *Bonmatin et al., 2015*). Secondly, neonicotinoids are water soluble and have a half-life in soil which can exceed 1,000 days (*Baskaran, Kookana & Naidu, 1999*; *De Cant & Barrett, 2010*; *Goulson, 2013*). When used as a seed dressing, only 1.6–20% of the active ingredient is taken up by the crop, the remainder leaching into soil water (*Sur & Storl, 2003*). Recent studies show
that concentrations of up to 9 ppb can be found in field margin plants near seed-treated crops (*Krupke et al., 2012*; *Stewart et al., 2014*). If these levels are typical, then we might predict considerable direct mortality in herbivorous insects feeding on field margin vegetation, as concentrations of 5–10 ppb are sufficient to control insect pests (*Castle et al., 2005*). The transportation of neonicotinoids by water courses means that butterflies feeding in other habitats could also be similarly affected.

Datasets investigating persistence of neonicotinoids within plants are very sparse. However, vines treated in spring maintain levels of imidacloprid sufficient to control pests throughout the growing season (*Byrne & Toscano, 2006*), and similarly levels of imidacloprid and thiamethoxam in citrus trees remained sufficient to suppress pests for 5 months (*Castle et al., 2005*). A single application of imidacloprid to maple trees protected them against insect pests for 4 years (*Oliver et al., 2010*). It is thus reasonable to propose that applications of neonicotinoids in adjacent crops could result in significant mortality of non-target invertebrates feeding in field margins for much of the spring and summer, and perhaps throughout the year. In the Netherlands, declines in insectivorous birds have been found to be associated with the amount of imidacloprid in surface water (*Hallmann et al., 2014*). This study concludes that the impacts of neonicotinoids on non-target invertebrates are causing the declines in insectivorous birds. Here we extend previous models (*Roy et al., 2001*) of the UK population indices of 17 species of widespread butterflies that commonly breed and forage on farmland with the addition of the number of hectares within the UK treated with neonicotinoid pesticides as a new explanatory variable.

## MATERIALS AND METHODS

### The data

We only selected widespread, resident species that routinely breed in any field or field margin habitats (Table 2), although all of the species also breed in other habitats. The indices for each species were derived from the UK Butterfly Monitoring Scheme (UKBMS) for the period 1984–2012 (*Brereton et al., 2011*). This dataset contains counts of butterflies from sites in a wide range of different habitats from across the UK. We included the full dataset for each species. The chosen period extends ten years before the introduction of neonicotinoid pesticides. Climatic data (total spring, total summer and total autumn rainfall, and mean spring and mean summer temperature) were obtained for the same period from the Met Office online database (*Met Office, 2014*). We used the climate data that has previously been found to affect UK butterfly populations (*Roy et al., 2001*) and which has been proposed to have played a role in the 2000–2009 decline (*Fox et al., 2011*). Usage of neonicotinoids in the UK (acetamiprid, clothianidin, imidacloprid, thiacloprid and thiamethoxam) was determined from *Defra*'s (*2014*) online PUS STATS database. A single total of the number of hectares treated with neonicotinoids was calculated for each year from 1994 (when they were first introduced) to 2012.

## Data analysis

The data were analysed using a linear mixed effect random slope model (*Gelman & Hill, 2007*) using the R 3.1.2 package lme4 (*R Project, 2015*). As the predictors were on very different scales all continuous predictors were transformed to Z-scores prior to analysis. These were used to model the butterfly indices for the period 1985–2012 for all 17 species combined using the following explanatory variables: mean spring temperature of the previous year, total spring rainfall in the previous year, mean summer temperature of the previous year, total summer rainfall in the previous year, total autumn rainfall in the previous year, mean spring temperature of the current year, total spring rainfall in the current year, mean summer temperature of the current year, total summer rainfall in the current year, year, population index for each species in the previous year, the number of hectares treated with neonicotinoids in the UK for the previous year. The previous year's climate data were included in the models as the adult population in a particular year may be influenced by the climatic conditions experienced by the previous (parent) generation during the breeding period and during the immature stages of their own lifecycle. Previous years' climate was found to be important in earlier models (*Roy et al., 2001*; *Brereton et al., 2011*). We used spring (March–May) and summer (June–August) weather as these are the main periods during which larvae and adults are active and were found to be important in previous models of UK butterfly population indices (*Roy et al., 2001*; *Brereton et al., 2011*). Autumn (September–November) rainfall in the previous year was included as an additional climatic variable as this is the period when neonicotinoids are generally used in the UK and rainfall is a likely mechanism by which they could be transported into the wider environment. Neonicotinoid usage the previous year was used for the same reason; most treated seeds are sown in autumn so it is the previous year's application which is most likely to affect the current year's butterfly population indices. Species was included as the only random effect. An interaction term between species and neonicotinoid usage the previous year was also included. All predictor variables were tested for collinearity, however none were considered to be correlated, with all Pearson correlation coefficients being less than 0.3. We used the dredge function of the MuMIn package (version 1.15.1) to identify the best models. The model coefficients and *P*-values were determined using the model.avg function of the MuMIn package including all models whose cumulative weight summed up to a total of 0.95. The relative contribution of each model was weighted by their relative weight score. After the minimum AIC mixed model was identified, random slope models including an interaction term between species and predictors maintained in the minimum AIC model were generated. The best model was again chosen using the dredge function. The ranef function was used to get the best linear unbiased predictions for the random effects.

Neonicotinoid use started in the mid-90s but the rapid expansion in their use occurred in the early part of the 21st century (*Defra, 2014*). Furthermore a recent study has suggested that concentrations need to reach a threshold level before negative effects occur on non-target species (*Hallmann et al., 2014*). Consequently the dataset was divided into two periods by year (1985–1998 and 1999–2012) to determine if there was a

**Table 1 Model of the impact of neonicotinoids on butterfly population indices.** The parameter estimates for the fixed effects included in the averaged linear mixed effect random slope models of butterfly indices for the 17 species, where species was included as a random effect and an interaction term was included between species and neonicotinoid usage the previous year. All other variables were included as fixed effects.

| Fixed effect | Parameter estimate | Standard error | T-value | P-value |
|---|---|---|---|---|
| Intercept | 3.597 | 4.518 | 0.795 | 0.425 |
| Population index (PY) | 0.067 | 0.008 | 8.36 | <0.001 |
| Neonicotinoid usage (PY) | −0.064 | 0.019 | 3.30 | <0.001 |
| Summer temperature | 0.064 | 0.008 | 8.30 | <0.001 |
| Spring rainfall (PY) | −0.026 | 0.007 | 3.58 | <0.001 |
| Summer rainfall (PY) | 0.020 | 0.009 | 2.13 | 0.033 |
| Spring temperature (PY) | −0.017 | 0.009 | −1.95 | 0.05 |
| Spring temperature | 0.014 | 0.008 | 1.88 | 0.061 |
| Spring rainfall | −0.010 | 0.008 | 1.31 | 0.191 |
| Summer temperature (PY) | −0.009 | 0.010 | 0.83 | 0.404 |
| Year | −0.002 | 0.003 | 0.70 | 0.485 |

**Notes.**

PY refers to previous year.

difference in the patterns of widespread butterfly indices before and after the introduction of neonicotinoids. The separation point was chosen as 1999 because 1998 was when neonicotinoid usage exceeded 100,000 hectares in the UK for the first time (and never dropped back below this level) which also resulted in each group containing the same number of years. A linear mixed effects model with year as a fixed effect and species as a random was applied to each half of the dataset.

## RESULTS

In keeping with previous studies, our linear mixed effect random slope model revealed strong positive associations between butterfly population index and both the previous year's index and summer temperature (Table 1). However, a strong negative association with the previous year's usage of neonicotinoids was also revealed. The pattern of association with neonicotinoid usage varied considerably between butterfly species (Fig. 1) but is associated with declines in most of the species (Table 2). The favoured habitat appears to be grassland for those species that exhibit the strongest negative associations with neonicotinoid usage.

A linear mixed effects model with year as a fixed effect and species as a random effect revealed that from 1985 to 1998 populations of these widespread butterflies actually exhibited a significant increase (parameter estimate = 0.0073, 95% CI [0.0029–0.0116]). The same model from 1999 to 2012 revealed a highly significant decline in butterfly populations (parameter estimate = −0.0145, 95% CI [−0.0187−−0.0102]). A linear mixed model including year and period as fixed effects and species as a random effect and an interaction term between year and period revealed that there was a significant difference

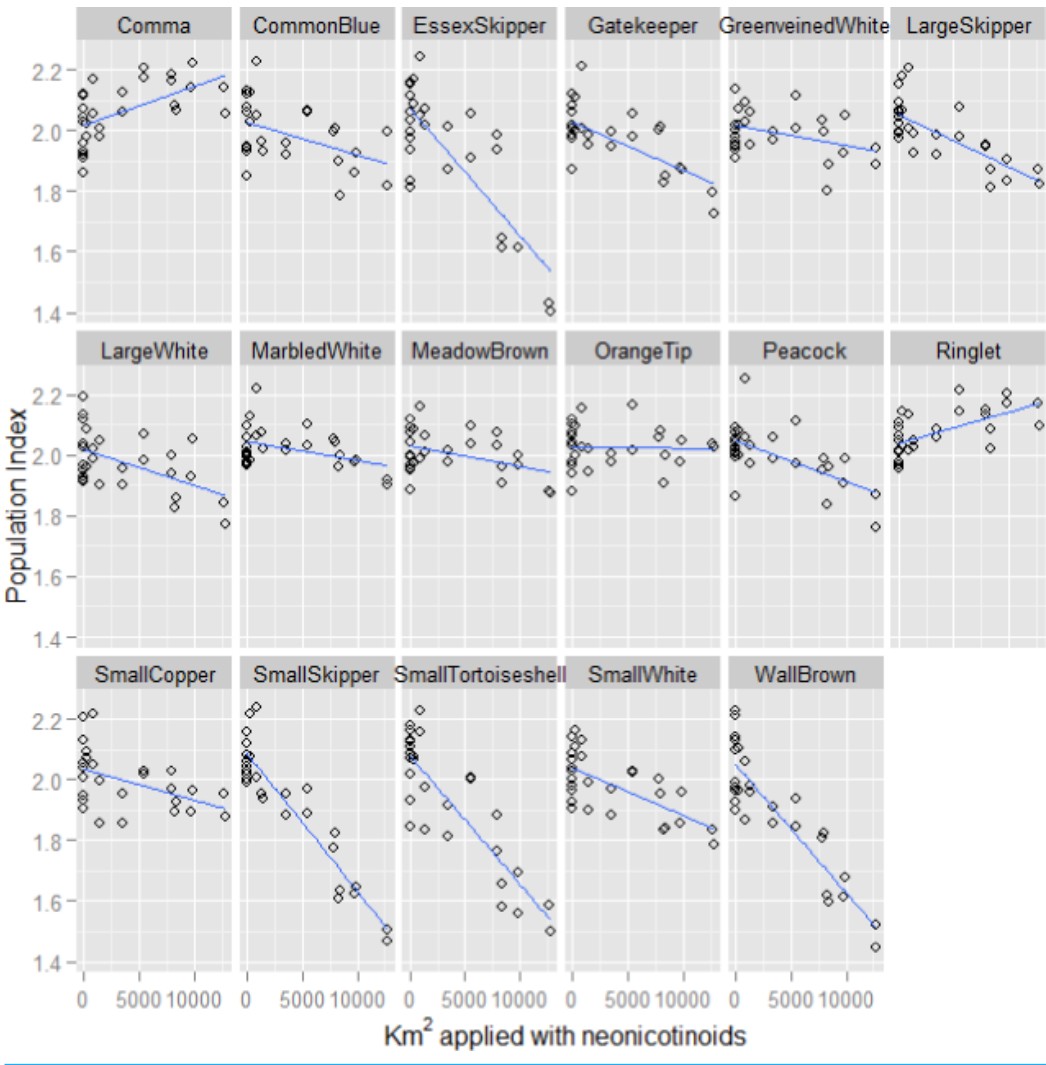

**Figure 1 Fitted values for each butterfly species plotted against neonicotinoid usage.** The fitted model values for the population indices for each species of butterfly plotted against the number of hectares applied with neonicotinoids the previous year from the linear mixed effect random slope model of butterfly indices, where species was included as a random effect and an interaction term was included between species and neonicotinoid usage the previous year. The index for each species is a log collated index scaled to have an average score of 2 across its entire time series (*Brereton et al., 2011*).

in the slope of change in butterfly population indices for these two periods (parameter estimate $= -0.0219$, 95% CI $[-0.0285--0.0147]$).

## DISCUSSION

The analysis carried out in this study extended previous models of UK butterfly population indices (*Roy et al., 2001*; *Warren et al., 2001*; *Brereton et al., 2011*) to include the number of hectares of UK farmland treated with neonicotinoid insecticides. A strong negative correlation was revealed between the populations of a group of widespread butterfly species characteristic of UK farmland and neonicotinoid usage. These findings are

**Table 2 Ten year trends and parameter estimates for the 17 butterfly species.** The UKBMS ten year population trends for 2000–2009 (*Fox et al., 2011*) and parameter estimates for each species from the averaged random slope linear mixed effect models for each of the 17 butterfly species, where species was included as a random effect and an interaction term was included between species and neonicotinoid usage the previous year. The habitat preferences for each species are also provided from *Oliver et al. (2009)* (G, grassland; HeMo, hedgerow and mosaic habitats; DW, deciduous woodland). The favoured habitat is given first and the number in parentheses denotes the proportion of the UK population found in that habitat (*Oliver et al., 2009*).

| Species | Effect of neonicotinoid usage on population index | 10 year population trend (2000–2009) | Habitat preference |
|---|---|---|---|
| Wall Brown, *Lasiommata megera* | −0.135 | −37% | G (0.45) HeMo DW |
| Small Skipper, *Thymelicus sylvestris* | −0.133 | −62% | G (0.53) HeMo DW |
| Essex Skipper, *Thymelicus lineola* | −0.131 | −67% | G (0.53) HeMo DW− |
| Small Tortoiseshell, *Aglais urticae* | −0.129 | −64% | G (0.49) HeMo DW |
| Gatekeeper, *Pyronia tithonus* | −0.069 | −23% | G (0.45) HeMo DW |
| Small White, *Pieris rapae* | −0.068 | −26% | G (0.40) HeMo DW |
| Large Skipper, *Ochlodes sylvanus* | −0.066 | −35% | G (0.46) HeMo DW |
| Large White, *Pieris brassicae* | −0.064 | −34% | HeMo (0.38) G DW |
| Common Blue, *Polyommatus icarus* | −0.061 | −30% | G (0.58) HeMo DW |
| Peacock, *Aglais io* | −0.058 | −24% | G (0.38) HeMo DW |
| Green-veined White, *Pieris napi* | −0.049 | −9% | HeMo (0.40) G DW |
| Small Copper, *Lycaena phlaeas* | −0.048 | −24% | G (0.54) HeMo DW |
| Meadow Brown, *Maniola jurtina* | −0.037 | −8% | G (0.55) HeMo DW |
| Marbled White, *Melanargia galathea* | −0.034 | −21% | G (0.60) HeMo DW |
| Orange-tip, *Anthocharis cardamines* | −0.022 | −8% | HeMo (0.39) G DW |
| Comma, *Polygonia c-album* | 0.007 | +34% | HeMo (0.42) DW G |
| Ringlet, *Aphantopus hyperantus* | 0.009 | +25% | G (0.42) HeMo DW |

correlative rather than causal and neonicotinoid usage may simply represent a proxy for other environmental factors associated with intensive agriculture. The parameter estimates from the model suggest that the strength of effect of neonicotinoids on butterfly population indices is equivalent to (albeit in the opposite direction from) the effect of the most important climatic variable, mean summer temperature. Neonicotinoid insecticide use has increased dramatically since it was first introduced to the UK in 1994, while the populations of many of the butterfly species have declined over the same period. Our model also included year as a fixed effect and revealed that, although the analysis is purely correlative, the recent declines in butterfly populations are much better explained by neonicotinoid usage than a linear decline over time. All 14 species which the model identified as being most negatively affected by neonicotinoid usage exhibited 10-year declines during the first decade of the 21st century (*Fox et al., 2011*) when neonicotinoid usage was increasing at its fastest rate (Table 2).

The explanatory strength of the neonicotinoid predictor in our models is all the more notable, and unexpected, as our focal group of 17 widespread butterflies typically found in arable farmland landscapes all occur in a wide variety of other habitats. The UKBMS population data used in our study are compiled from sites representing many different

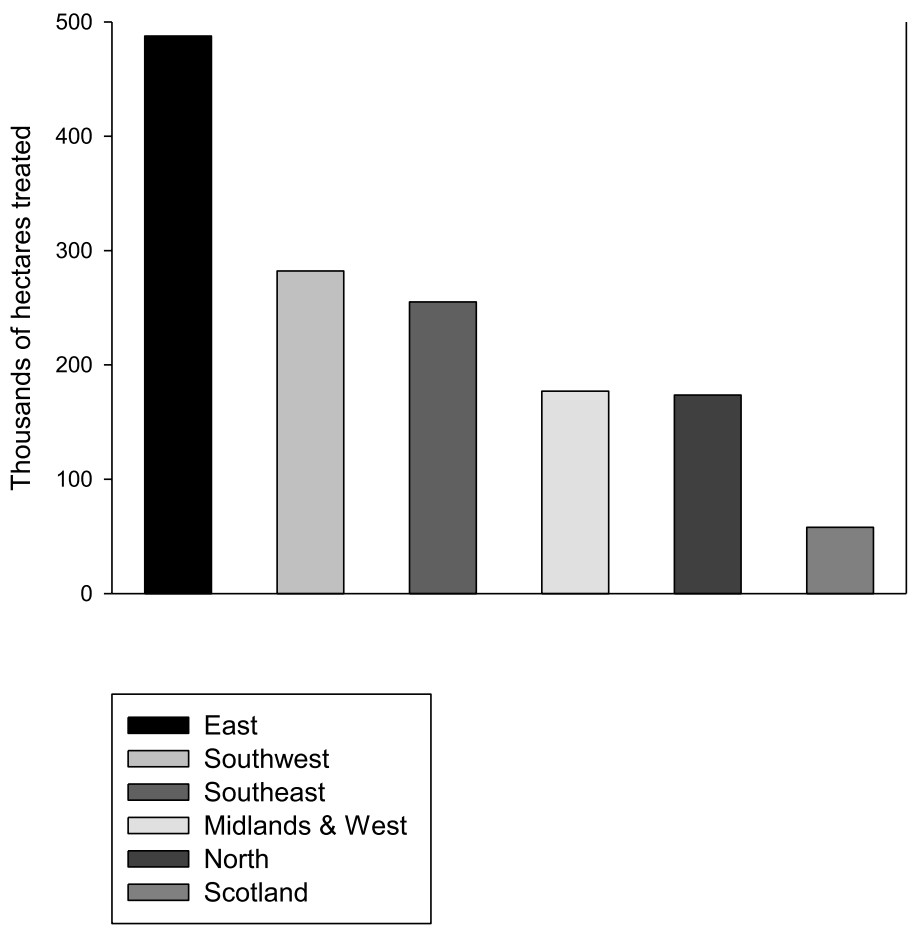

**Figure 2** **Neonicotinoid usage by region.** The area in thousands of hectares treated with neonicotinoids in 2010 in different regions of England and Scotland. Data from *Defra (2014)*.

habitats. Indeed, UKBMS sites usually comprise semi-natural habitats managed, at least in part, for biodiversity conservation. Thus, the strong correlation between neonicotinoid usage and national butterfly trends is remarkable given that relatively few of the monitored butterfly populations would be from arable farmland.

Neonicotinoid usage, which represents a recent change to the environment, may explain the concurrent rapid decline in butterfly populations. Interestingly, a study of patterns of declines across the UK butterflies found that generalist butterflies are not declining in Scotland (*Brereton et al., 2011*) where neonicotinoid usage is much lower than it is in England (Fig. 2; *Defra, 2014*). In the Netherlands, where neonicotinoids are also widely used, major declines in populations of widespread butterflies have also been observed (*Van Dyck et al., 2009*). Furthermore, the spatial pattern of decline in insectivorous birds in the Netherlands correlates significantly with the presence of neonicotinoids in the environment (*Hallmann et al., 2014*). For some butterfly species the observed effects may partly be due to adults being exposed while nectaring on treated oilseed rape, the main flowering crop in the UK that is treated with neonicotinoids. However, oilseed rape flowers in April–May, before the adults of other species seemingly affected by neonicotinoids are

on the wing. It seems probable then that any adverse effects of neonicotinoids would be mediated primarily by contamination of larval food plants in the wider environment such as field margins, although the declines in *Pieris brassicae* and *P. rapae* may also be explained by the fact that they lay some of their eggs on the oilseed rape itself.

Habitat deterioration is the main candidate proposed to explain long-term declines in distribution and abundance of butterflies in the UK during the 20th century (*Warren et al., 2001*) but there is no clear evidence that habitat has continued to deteriorate in the 21st century to the extent that it can explain the sudden crash in populations of widespread species after a period of stability during the latter part of the 20th century, so this hypothesis is difficult to test. Habitat deterioration is difficult to quantify at large geographic scales and, therefore, has not been included in previous models of butterfly trends (*Roy et al., 2001*; *Warren et al., 2001*; *Brereton et al., 2011*). If habitat deterioration is the main cause of butterfly declines and agricultural intensification is playing a key role in the loss of habitat, then neonicotinoid usage might be acting as a proxy for agricultural intensification and therefore habitat deterioration in our models. Thus, neonicotinoid usage could be responsible for driving butterfly declines or alternatively it could provide the first useful quantifiable measure of agricultural intensification that strongly correlates with butterfly population trends.

It should be noted that most UKBMS sites have not been characterised by habitat although many comprise semi-natural habitats of relatively good quality compared to arable farmland and thus may not be directly exposed to neonicotinoids. However butterflies are mobile organisms and declines at UKBMS sites could occur if they are surrounded by farmland populations that act as population sinks (*Dias, 1996*). Furthermore, contamination of water by neonicotinoids, which correlates strongly with the decline in insectivorous birds in the Netherlands (*Hallmann et al., 2014*), provides the potential for rapid transport of neonicotinoids from arable farmland to other surrounding habitats. This means that most other impacts of agricultural intensification are likely to act on a relatively localised scale compared to neonicotinoids. As most UKBMS sites are not in arable farmland, this favours the hypothesis that neonicotinoids are directly driving the declines in butterflies rather than acting as a proxy for agricultural intensification.

Three areas of further research are required to elucidate the correlation we have found between neonicotinoid use and butterfly populations. Experimental studies to determine the toxicity of neonicotinoids to butterflies are required as a matter of utmost urgency. Improved understanding of the levels of neonicotinoid contamination in field margin plants and surface water are also needed to assess potential exposure. Finally, even if widespread butterflies are being routinely exposed to harmful levels of neonicotinoids, further evidence is required to determine if neonicotinoids are directly responsible for declines in national butterfly populations or are acting as a proxy for other factors associated with agricultural intensification. Further development of the UKBMS models to incorporate land cover and habitat descriptors, as well as additional climatic variables could add further insight into the factors playing a role in the recent declines. If neonicotinoids are driving the decline in widespread butterflies in the UK, this begs the

questions as to what other non-target arthropods might be similarly affected and whether neocotinoids are playing a role in the declines in insectivorous farmland birds in the UK.

## ACKNOWLEDGEMENTS

The authors are indebted to all the UKBMS volunteer recorders for collecting data, and are grateful to three reviewers of the manuscript for their constructive and helpful comments.

### Funding

The authors received no funding for this work.

### Competing Interests

The authors declare there are no competing interests.

### Author Contributions

- Andre S. Gilburn conceived and designed the study, analyzed the data, wrote the paper, prepared figures and/or tables, reviewed drafts of the paper.
- Nils Bunnefeld analyzed the data, reviewed drafts of the paper.
- John McVean Wilson analyzed the data, reviewed drafts of the paper, set up the dataset from other databases.
- Marc S. Botham, Tom M. Brereton and Richard Fox oversaw and provided the UKBMS dataset, wrote the paper, reviewed drafts of the paper.
- Dave Goulson conceived and designed the study, wrote the paper, reviewed drafts of the paper.

### Supplemental Information

Supplemental information for this article can be found online at http://dx.doi.org/10.7717/peerj.1402#supplemental-information.

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
