# Peer review of "Are neonicotinoid insecticides driving declines of widespread butterflies?"

_PeerJ, doi:10.7717/peerj.1402_

## Round 0.1 · original submission · Minor Revisions

Please make revisions according to the reviewers' comments.

·

Basic reporting

No comments.

Experimental design

No comments.

Validity of the findings

No comments.

Additional comments

The authors are to be commended for being totally up-front in disclaiming any proof of causation and in specifically noting potentially confounding variables. This is especially important because inevitably this paper will be seized upon by environmental "activists" and the media and represented as presenting a more definitive association than it does. That will NOT be any fault of the authors or of the journal, but it is virtually certain to occur.

Reviewer 2 ·

Basic reporting

The authors report a very interesting correlative relationship between the use of neonicotinoid insecticides and population trends of 17 widespread butterfly species in Britain. They show that the decline of the common butterfly species coincides with the expansion of the use of neonicotinoids, and that the variable describing the annual amount of used neonicotinoids significantly adds explanatory power to the statistical model when all the other previously identified important weather and population variables are included. Although no causal relationship is shown, the observed patterns shown together with what is previously reported on neonicotinoids suggest that neonicotinoids might be a significant factor behind the recently seen substantial declines of widespread butterflies. This is a novel and widely interesting result, and very likely to inspire more research on this topic.

The manuscript is concise and well-written. I only have a few minor comments concerning the basic reporting:

1. Line 57: Pisa et al. 2015 is missing from the list of references.

2. Lines 63-65: I suppose that here it would be appropriate also to cite the recent paper by Rundlöf et al. reporting negative effects of neonicotinoids on wild bees in field conditions (Rundlöf et al. 2015: Seed coating with a neonicotinoid insecticide negatively affects wild bees, Nature 521:77–80)?

3. Line 314: Please add the information on the journal volume and page numbers.

4. Line 350: Should be “Environmental”.

5. Table 2: Add italics also to Ochlodes sylvanus.

6. Fig. 1: The numbers in the x-axis are hardly readable. What is the interpretation of the Population index in the y-axis? Does index = 1 refer to the level at the beginning of the monitoring (or maybe something else, because the values are always higher than 1)?

Experimental design

The research question is interesting and has been clearly defined. The results are based on an appropriate large data set and adequate, well-designed statistical analyses.

Validity of the findings

I would suggest two small additional considerations to complement the interpretation of the results:

1. The authors point out that the butterfly species most commonly occurring in field margins are probably the most vulnerable to the negative effects of neonicotinoids. Thus it would be interesting to include some information on the habitat (especially field margin) use of the 17 focal butterfly species (potentially adding a new column in Table 2). Although all these species are sometimes seen in field margins, there is probably quite a lot of variation between the species, and it would be useful to document it. Probably such information is available in Britain (e.g. via the wider countryside butterfly survey; in addition Oliver et al. 2009, Ecology Letters 12:1091–1102, has also published some quantitative habitat use data that might be useful here). If suitable data would be available on habitat use, then it would be interesting to check whether the species most commonly occurring in field margins have higher modelled effect of neonicotinoid usage on the population index. Maybe this kind of additional analysis could also explain the differing behaviour of the two species, the comma and the ringlet, in the statistical analyses compared to the 15 other widespread species?

2. In Fig. 2 it would also be useful to show the percentages of all cultivated fields that are treated with neonicotinoids in the different geographical areas. If the percentages are high in some areas, then a convincing direct effect of neonicotinoids on butterflies would seem possible. However, if the percentages are always low, then the alternative potential explanation offered by the authors would seem more probable. This might be a rather essential issue worth commenting in the Discussion.

Additional comments

I included my general comments already above in the Basic Reporting.

·

Basic reporting

No comments

Experimental design

No Comments

Validity of the findings

No Comments

Additional comments

Select the drivers you want to blame? …this was my first thought when I was asked to review to paper by Gilburn et al. – but after having read the abstract my interest grew and I studied the manuscript with high expectations.

To start with my overall judgement: I very much favour that this paper gets published, especially as already early on the authors state that “further research is needed urgently to show whether there is a causal link between neonicotinoid usage and the decline of widespread butterflies or whether it simply represents a proxy for other environmental factors associated with intensive agriculture”.

Generally I find it difficult to use the total area on which neonicotinoids were applied on a very coarse level (5 regions of England, and Scotland as a 6th region) as a predictor variable – a higher level of differentiation might have been more appropriate (or are these not available?), given that butterfly monitoring transects are well distributed across the UK. Also a more differentiated and spatially more explicit selection and treatment of climate variables might have been advisable. One of the authors of the present paper has shown how this can be done – Tom Brereton who is also co-author of the newly published paper of Oliver et al. (2015). There drought was identified as an important explanatory climate variable – with very different drought recoveries of 6 selected butterfly species (5 of them also part of the 17 species dealt with here: P. brassicae, P. rapae, P. napi, O. sylvanus, A hyperanthus). The latter very much depends upon landscape structure and fragmentation. In the present study the correlation of the most important climatic variable, mean summer temperature, was a positive one – and as strong as the negative one of neonicotinoids. Oliver et al. (2015) present a good example of how a “slightly” different basic hypothesis might lead to the selection of a different level of detail and thus explanatory variables (while using the same very admirable dataset of the British Butterfly Monitoring Scheme as did Gilburn et al).

Surprisingly, the authors tend to argue “that the fact that most UKBMS sites are not in arable farmland actually favours the hypothesis that neonicotinoids are directly driving the declines in butterflies rather than acting as a proxy for agricultural intensification”. Here I find it really hard to follow, as from my point of view the low proximity rather raises doubts regarding the role of neonics as direct driver – and because of this I even would question their role as proxy for agricultural intensification. On the other hand, why didn’t the authors use the increase of the two species Polygonia c-album and Aphantopus hyperantus, which are surely no typical representatives of the open agricultural landscape, to argue for less pronounced neonic effects in non-arable areas?

Nevertheless, I think it is important to make this paper available to the scientific and wider community in order to have a broader discussion and to think along different lines. I very much agree with the authors, that further research is required to elucidate the correlations they have found.

Reference cited:
Oliver TH, Marshall HH, Morecroft MD, Brereton T, Prudhomme C, Huntingford C. (2015). Interacting effects of climate change and habitat fragmentation on drought-sensitive butterflies. Nature Climate Change 5; published online: 10 August 2015; DOI: 10.1038/NCLIMATE2746

---

## Round 0.2 · accepted · Accept

Thank you for your careful revisions.